# All-optical switching of an epsilon-near-zero plasmon resonance in indium tin oxide

Justus Bohn [1✉], Ting Shan Luk[2,3], Craig Tollerton[1], Sam W. Hutchings[1], Igal Brener [2,3], Simon Horsley [1], William L. Barnes [1] & Euan Hendry[1]

Nonlinear optical devices and their implementation into modern nanophotonic architectures are constrained by their usually moderate nonlinear response. Recently, epsilon-near-zero (ENZ) materials have been found to have a strong optical nonlinearity, which can be enhanced through the use of cavities or nano-structuring. Here, we study the pump dependent properties of the plasmon resonance in the ENZ region in a thin layer of indium tin oxide (ITO). Exciting this mode using the Kretschmann-Raether configuration, we study reflection switching properties of a 60 nm layer close to the resonant plasmon frequency. We demonstrate a thermal switching mechanism, which results in a shift in the plasmon resonance frequency of 20 THz for a TM pump intensity of 70 GW cm$^{-2}$. For degenerate pump and probe frequencies, we highlight an additional two-beam coupling contribution, not previously isolated in ENZ nonlinear optics studies, which leads to an overall pump induced change in reflection from 1% to 45%.

[1] School of Physics, University of Exeter, Exeter, UK. [2] Sandia National Laboratories, Albuquerque, NM, USA. [3] Center for Integrated Nanotechnologies, Sandia National Laboratories, Albuquerque, NM, USA. ✉email: jb933@exeter.ac.uk

Nonlinear optics is utilized for a wide range of photonic applications such as quantum all-optical data processing[1,2], information technology[3,4] and tele-communication applications. With the rise of new computational demands such as artificial intelligence, all-optical signal processing is often seen as a breakthrough technology for the next generation of computation and communication devices[5]. However, such applications are limited by the interaction of light signals, with extremely weak optical nonlinearity exhibited by most materials. This leads to high power consumption and a large physical size of optical circuitry, making intsegration into existing nanophotonic platforms challenging[6,7]. Moreover, most optical switching materials and geometries are not compatible with existing complementary metal-oxide-semiconductor fabrication technologies, which is preferential for implementation into existing platforms[8].

Recently, epsilon-near-zero (ENZ) materials have attracted much attention, not only for their intriguing linear properties[9] but also because they exhibit large optical nonlinearities[10,11]. Moreover, a subset of ENZ materials, transparent conductive oxides, exhibit resonance frequencies in the near-infrared, thereby offering the potential for integrated telecom applications[12–17]. Indium tin oxide (ITO), as one example, has been shown to undergo a refractive index change of order unity upon optical pumping of a thin film[18]. This effect is thought to arise from electron heating, which leads to a change in effective mass due to the non-parabolic electron dispersion[19,20]. Similar optical nonlinearities have been measured for doped zinc oxides[21,22] and CdO[23]. These materials are also tunable, with variability in their doping level, giving control over the ENZ resonance wavelength spanning the infrared range[24,25]. To further increase the optical switching properties of transparent conducting oxides, different strategies have been employed, including additional structuring[16,26,27] or the design of cavity modes[23,28–30].

It is also well known that near the ENZ frequency of a thin transparent conducting oxide one can also excite a plasmon resonance, giving rise to enhancement of the incoming field and near-perfect absorption[31]. We will refer to this resonance as the ENZ plasmon. One can excite plasmons using a high index incident prism in the Kretschmann–Raether configuration, circumventing the need for nano-structuring or the additional support of a cavity. This approach has been employed to study plasmon-based nonlinear optical dynamics in gold films[32–34]. However, while ENZ plasmon excitation has been employed to enhance third harmonic radiation[35], there has been no study of all-optical switching in transparent conducting oxides in this desirable geometry.

Here, we investigate optically induced shifting of the ENZ plasmon frequency via pump–probe experiments in the Kretschmann–Raether configuration. This geometry provides a potential switching platform from near-total absorption to total internal reflection upon tuning the plasmon resonance into and out of the spectral range. We identify two contributions to the nonlinear signal: A dominant thermal switching process results in a shift in the plasmon resonance frequency of 20 THz for a TM pump intensity of 70 GW cm$^{-2}$ when pumping resonantly, resulting in a change in reflection of the probe from 1 to 30%. Exclusively for the TM pump polarization, we isolate an additional two-beam coupling (TBC) contribution. These two mechanisms combine to enable reflection switching of a 60-nm layer by more than an order of magnitude, with a measured change in reflection of the probe from 1 to 45% for a pump intensity of 70 GW cm$^{-2}$.

## Results

Wavevector matching is required to excite a plasmon. In our design, we follow the Kretschmann–Raether configuration as seen in Fig. 1a. The prism enables the pump and probe beams to be incident upon the ITO layer beyond the critical angle, resulting in a wavevector beyond the air light line and thereby enabling plasmon excitation in the near-perfect absorption regime. Of interest, here is the area around epsilon being zero, which coincides with the backbended segment of the plasmon dispersion, shown in Fig. 1c for a 60-nm-thick film (dispersion model found in the Supplementary Note S2). We refer to this plasmon segment in the remainder of this paper as the ENZ plasmon region. We are specifically interested in the non-radiative ENZ plasmon, which lies beyond the air light line—in this region one expects to observe a near-perfect absorption and enhanced fields[31].

Recent measurements have shown that pumping below the bandgap of a transparent conducting oxide leads to carrier heating, which subsequently results in an increased effective mass and decreased plasma frequency[26]. Such an intensity-dependent plasma frequency should materialize as a shifting resonance frequency of an ENZ plasmon. We study this effect in the ENZ plasmon dispersion of a 60-nm ITO thin film using a pump–probe scheme. To begin, we pump with TE polarization and probe with TM polarization—this removes coherent interference of pump and probe pulses at the cost of less efficient absorption. Figure 2a shows three typical pump–probe measurements where we pump and probe different regions of the plasmon resonance. For a probe frequency of 240 THz (red, case I) the ENZ resonance redshifts away from the probe, the absorption decreases and we see an

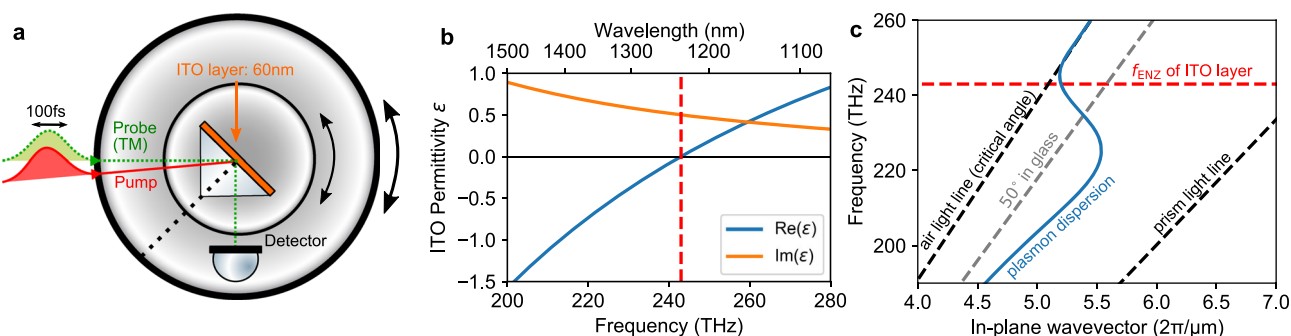

**Fig. 1 Basic material properties and schematic set-up. a** Schematic pump–probe set-up with the ITO sample index matched to a prism in order to probe the ENZ plasmon beyond the critical angle (here, $\theta = \theta_{pr} = 45°$). In this geometry, the relative beam angles are fixed such that the pump angle is always given as $\theta_{pm} = \theta_{pr} - 3.4°$. **b** Optical permittivity of the ITO film used in this study, measured using ellipsometry, with an epsilon-near-zero frequency of 243 THz (red dashed). **c** Plasmon dispersion branch of the 60-nm ITO thin film closest to the air light line (blue), calculated following Supplementary Note S3.

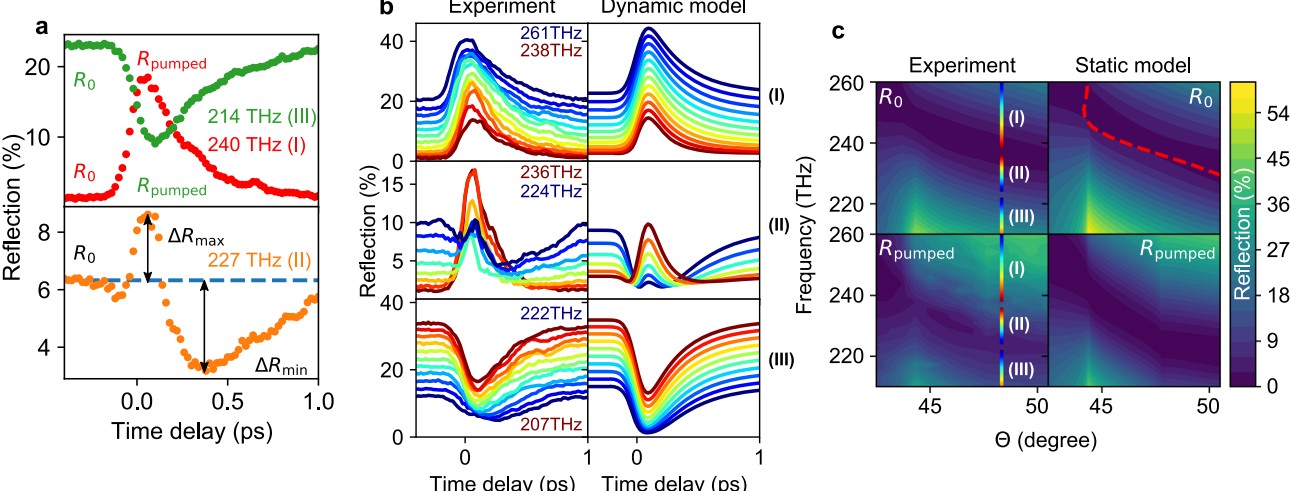

**Fig. 2 Time dependence of the nonlinear reflection for degenerate pump and probe frequencies. a** Three typical pump–probe measurements, where: (I) the probe frequency (240 THz, red) is larger than the plasmon resonance frequency, (II) the resonance shifts spectrally through the probe during pumping (227 THz, orange), and (III) the probe frequency is smaller than the pumped resonance (214 THz, green). The probe is TM polarized to study the plasmon response, while the 70 GW cm$^{-2}$ pump is TE polarized to avoid coherent contributions. We signify the geometry and frequencies using the notation for the probe (pr) and pump (pm) following (pr: TM, 48.3°, $f_{pr}$ = 240, 227, 214 THz; pm: TE, 44.9°, $f_{pm} = f_{pr}$). **b** Various pump–probe measurements divided into the previously discussed three case types. The dynamic model shows the reflection coefficient for a time-varying effective medium (pr: TM, 48.3°, $f_{pr}$ = 261...207 THz; pm: TE, 44.9°, $f_{pm} = f_{pr}$). **c** Scans of the probe reflection over incoming angle and degenerate frequency, showing the initial $R_0$ (top) and the pumped case $R_{pumped}$ (bottom). The vertical jet coloured lines at 48.3° indicate the measurements presented in panel (**b**). The transfer matrix model on the right shows the expected initial reflection at the top with the ENZ plasmon dispersion from Fig. 1c (red dashed) with $R_{pumped}$ based on the carrier heating nonlinearity at the bottom (modelling details in S3) (pr: TM, 42.5...50.5°, $f_{pr}$ = 261...210 THz; pm: TE, 39.1...47.1°, $f_{pm} = f_{pr}$).

increase in reflectivity. For 214 THz (green, case III), the ENZ resonance shifts spectrally towards the probe, the absorption increases and we see a decrease in reflectivity. Both cases display expected temporal dynamics: a fast, ~100 fs (pulse limited) initial change, followed by a slower, ~1 ps thermal relaxation, similar to the previous studies[18,21,36].

To better describe these effects, we have introduced a dynamic model, which captures effects arising due to rapid time-dependent changes to the reflection coefficient (details in Supplementary Note S3).

For 227 THz (orange, case II), the ENZ resonance shifts spectrally through the probe, and we observe some rather unusual dynamics in the pump–probe signal that, to the best of our knowledge, have not previously been observed and discussed. For this case, one initially observes a decrease in reflection, followed by an increase in reflection, as heating causes the resonance to shift through the probe frequency. Subsequent cooling then returns the resonance to its starting frequency. These combined effects cause multiple oscillations in the pump–probe dynamics. To better describe these effects, we have introduced a dynamic model, which captures effects arising due to rapid time-dependent changes to the reflection coefficient (details in Supplementary Note S3). With this model, we can reproduce the features seen in our pump–probe signals: the three broad types of behaviour are shown for various frequencies in Fig. 2b and modelled using time-varying reflection coefficient implied directly from experimental observations. For the first two cases, it is straightforward to define maxima/minima in reflection ($R_{pumped}$) relative to the initial reflection ($R_0$), as labelled in Fig. 2a. Defining a maximal response in the oscillatory case is problematic—for simplicity, we define $R_{pumped}$ = $R_0 + \Delta R_{max} + \Delta R_{min}$ for all measured time delay scans. While not being able to resolve the transient features seen in Fig. 2b, this reduces the complex dynamical information contained in one complete time delay scan into two quantities of interest: the reflection before arrival of the pump ($R_0$) and a quantity describing the maximally changing reflection on photoexcitation

($R_{pumped}$). This allows us to condense information from many measurements into one colour plot.

By varying both the incident angle and the degenerate pump/ probe frequency, one can observe the plasmon dispersion shift (Fig. 2c). Upon illuminating the sample with a TE pump of 70 GW cm$^{-2}$ intensity, we measure a redshift of the ENZ plasmon resonance, as seen for the pumped reflection ($R_{pumped}$) relative to the initial reflection ($R_0$). We interpret these changes using a simplified "static" model, which calculates the reflection using a transfer matrix model assuming an effective medium with an intensity-dependent permittivity for the ITO layer (details in Supplementary Note S3, static model). We assume a linear intensity-dependent shift of the plasma frequency, $\omega_p(I) = (1 + \omega_{p,2}I)\,\omega_{p,0}$, where $I$ is the calculated absorbed intensity and $\omega_{p,2}$ is the nonlinear fit parameter. This approach gives good agreement for the red-shifting behaviour of the resonance, while also confirming some more subtle features, such as the critical angle feature for the probe pulse just below 45°, as well as a second critical angle feature near 48°, which arises due to the 3.4° difference in angle between pump and probe. We find best agreement with the data for an intensity-dependent red-shifting of the plasma frequency, described by $\omega_{p,2} = -0.38\%$ cm$^2$ GW$^{-1}$. This is the general behaviour expected for heating of the electron plasma in ITO, an effect that arises due to the non-parabolicity of the conduction band in this material[18,23,37]. However, as discussed in Supplementary Note S3, comparison between experiment and modelling suggests that the electron heating is weaker in our ITO compared to that previously reported in ref. [18], as our extracted value for $\omega_{p,2}$ is similar despite an increased local intensity arising due to the Kretschman geometry. We do not fully understand this discrepancy, but it may arise due to complications in the homogeneous analysis used in ref. [18] (discussed in detail in the Supplementary Note S3), or due to variations between ITO samples. As further discussed in the Supplementary Note S3, we cannot identify changes to the scattering rate as readily as the changes to the plasma frequency, as the scattering

rate affects mainly the width of the plasmon resonance, which is impacted by an artefact of our analysis using our static model (discussed below). However, it is expected that heating should have minimal effect on the scattering rate in transparent conducting oxides due to the dominance of impurity scattering[20]. As shown explicitly in Supplementary Fig. S2, and discussed in the surrounding Supplementary Note S3, comparison between our dynamic model and experimental data suggests that pump induced changes to scattering rate are negligible.

In Fig. 3, we vary pump intensity while fixing the pump frequency to 240 THz and incident angle to just beyond the critical angle of the pump (48.3°). The pump polarization is TE, that is, non-resonant with the ENZ plasmon. On increasing pump intensity, we observe a clear redshift of the ENZ plasmon, with approximately linear intensity dependence. We also observe a more subtle effect: an apparent slight narrowing of the resonance. This is an artefact of our analysis, arising from the oscillatory features for case II, which become more prominent for increasing intensity. Since we are unable to remove completely, these effects

in our static analysis, they give rise to a slight distortion of the resonance lineshape for high intensities.

Finally, we aim to maximize switching: by exciting with a TM pump, a more efficient energy deposition via the ENZ plasmon resonance is expected. We investigate the nondegenerate frequency dependence of the nonlinear material response by fixing the external pump intensity to 70 GW cm$^{-2}$ and incident pump angle to 48.3°, and compare TE and TM excitation. First, the pump absorption for TE excitation is only weakly frequency dependent. For this reason, we observe a pump frequency-independent shift of the ENZ resonance of 14 THz in Fig. 4a. For TM excitation, seen in Fig. 4b, we see two additional interesting aspects. For on-diagonal (degenerate) measurements, we can identify a noticeably larger reflection due to TBC. We note that the TBC contribution may be concealed, and not readily separated, within other degenerate pump–probe measurements in the literature. In Fig. 4c we compare the coherent and incoherent contributions to our signal. For the nondegenerate case, we observe thermal switching behaviour resulting in a large change in reflection from $R_0 \sim 1\%$ to $R_{pumped} \sim 30\%$. This corresponds to a shift of the plasmon resonance frequency of 20 THz, which is more than four times the spectral width of a 100 fs pulse. In the degenerate case, we see a further increase of the differential reflection to $\Delta R \sim 45\%$ due to the interference between the pump and probe beams (details in Supplementary Note S4). Only by systematically varying both pump and probe frequencies independently have we been able to identify this coherent contribution to switching.

In conclusion, we present an investigation into all-optical switching of ENZ plasmons. We identify two contributions to our nonlinear signal: a thermally driven switching process results in a shift in the plasmon resonance frequency of 20 THz for a pump intensity of 70 GW cm$^{-2}$, while additional TBC is observed for degenerate pump and probe frequencies. In total, we observe switching of more than one order of magnitude, from $R_0 \sim 1\%$ to $R_{pumped} \sim 45\%$, resulting entirely from resonant conditions allowed by the geometry. For comparison, switching of ITO in air with 70 GW cm$^{-2}$ has been shown to result in a change in transmission from 12 to 34%[18], for samples with five times the material thickness than those studied here. Considering the 16.5 dB extinction ratio achieved in this proof-of-principle study, switching from near-perfect absorption to high reflection is useful from a signal processing point of view, and could pave the way towards optical plasmon switching at telecom frequencies.

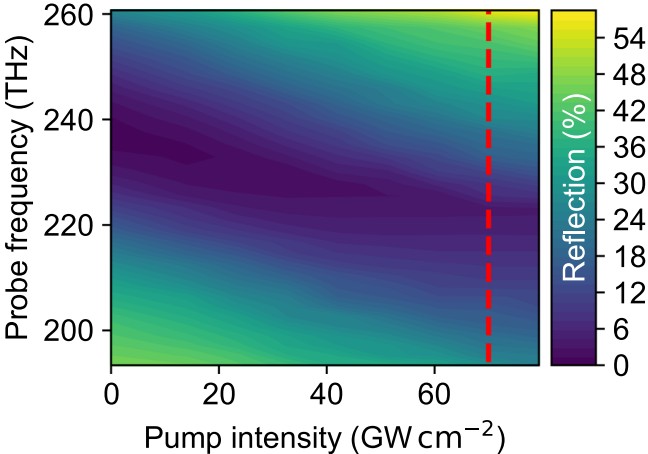

**Fig. 3 Intensity dependence.** Intensity-dependent $R_{pumped}$ for a constant incident probe angle of $\Theta = 48.3°$ and a pump with a constant frequency of 240 THz (TE polarized, i.e. non-resonant with the ENZ plasmon). The ENZ plasmon redshifts with increasing pump intensity. The red dashed line indicates the 70 GW cm$^{-2}$ used for the measurements in Fig. 2. (pr: TM, 48.3°, $f_{pr}$ = 260...190 THz; pm: TE, 44.9°, $f_{pm}$ = 240 THz).

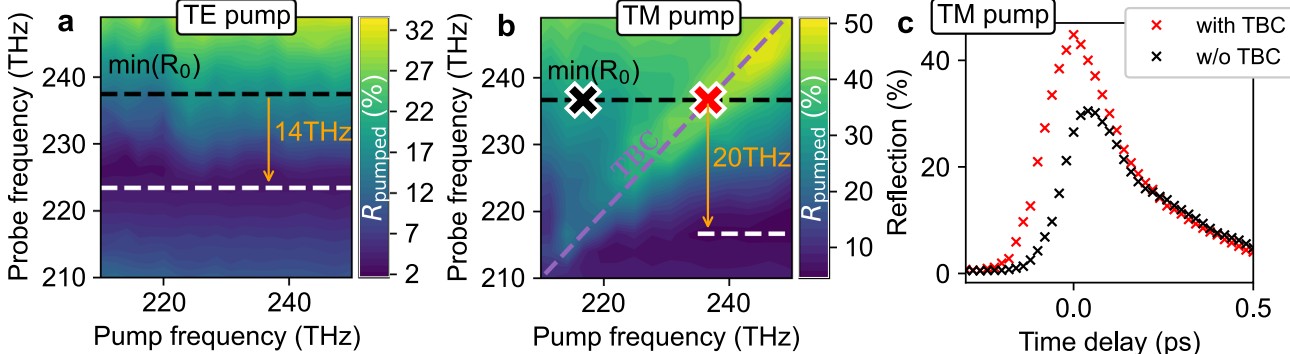

**Fig. 4 Nondegenerate frequency dependence and coherent contribution.** We examine the pump–probe frequency dependence of $R_{pumped}$ for 70 GW cm$^{-2}$. **a** Using a TE pump, the frequency shift of the resonance is 14 THz, and independent of pump frequency. The dashed lines indicate the initial (black) and pumped (white) ENZ plasmon resonance frequency (pr: TM, 48.3°, $f_{pr}$ = 250...210 THz; pm: TE, 44.9°, $f_{pm}$ = 250...210 THz). **b** A TM pump highlights the two-beam coupling (TBC) for equivalent polarization and frequency. The higher pump absorption through the plasmon resonance leads to a larger, 20 THz shift of the resonance frequency (pr: TM, 48.3°, $f_{pr}$ = 250...210 THz; pm: TM, 44.9°, $f_{pm}$ = 250...210 THz). Markers indicate the time delay scans compared in **c**, where the probe frequency is 237 THz, while the pump frequency is either 237 THz (red, with TBC) or 217 THz (black, without TBC) (pr: TM, 48.3°, $f_{pr}$ = 237 THz; pm: TM, 44.9°, $f_{pm}$ = 237, 217 THz).

Currently achievable switching rates are limited by further experimental requirement such as the need for high-pulse-energy femtosecond laser amplifiers. While the switching performance of the layer and the requirement for high intensities could be further improved by more elaborate layer designs and/or further material improvements such as in carrier mobility, achievable switching rates will be ultimately restricted by limitations in repetition rate of pulsed optical sources. Finally, the compatibility with complementary metal-oxide-semiconductor fabrication technique makes thin TCO layers and their ENZ plasmon feature a compelling route for nonlinear integrated-photonics applications without the need for nanostructure or building additional cavities, while better matching the spatial modes used in photonic circuit systems.

## Methods

**Numerical simulation and calculation**. The plasmon dispersion was calculated using Mathematica 12. The static and dynamic model for the three-layer system were calculated using Python 3.7. Details of both models can be found in the Supplementary information.

**Sample fabrication**. ITO was sputtered onto cover glass at room temperature using 90/10 $In_2O_3$/$SnO_2$ Kurt Lesker target and sputtering tool. The base pressure before deposition was in low $10^{-6}$ torr, but raised to 3 mT of Ar only during deposition with an RF power of 145 W. In order to achieve high carrier density, both deposition and annealing was performed under a lowest possible residual oxygen environment. The samples are post-annealed in forming gas for 3 min at temperatures between 425 and 525 °C.

**Optical set-up**. For the pump–probe measurements, we used an amplified Ti: sapphire laser (Legend Elite, Coherent), with a central wavelength of 800 nm, pulse duration of 107 fs and repetition rate of 1 kHz, feeding two identical OPAs (TOPAS, Light Conversion). The signal output of one OPA was used as the pump, and the signal output of the other OPA was used as the probe, allowing us independent control of pump and probe frequencies. The pump was focused using a 40-cm BK7 lens, the probe with a 25-cm $CaFl_2$ lens. The pump beam diameter (full-width at half-maximum) was measured to be 800 µm in air, while the probe was 250 µm. To make sure the probe intensity is significantly smaller than the pump, we used several additional OD filters to decrease the probe power and tested that the nonlinear reflection was independent of adding/removing filters. We used a referenced tuneable filter wheel to enable a frequency-independent pump power. The probe beam was chopped and measured with a fibre connected detector combined with a boxcar and lock-in electronic set-up. The ITO thin layer was positioned on top of a 180 µm coverslip and attached to the front of a right-angle prism (EKSMA, UV-FS 5 × 5 mm²) by applying an index matching fluid (Olympus IMMOIL-F30CC). To obtain absolute reflection measurement, we used the total internal reflection of a blank coverslip as a reference. The angle of incidence of the pump was set to be ~3.4° smaller than that of the probe (5° in air).

## Data availability

The datasets generated during and/or analysed during the current study are available from the University of Exeter's institutional repository at: https://doi.org/10.24378/exe.3004. The Python and Mathematica codes to analyse and plot the data are included.

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

## Acknowledgements

We acknowledge the financial support from the Engineering and Physical Sciences Research Council (EPSRC) of the United Kingdom, via the EPSRC Centre for Doctoral

Training in Metamaterials (Grant No. EP/L015331/1). SARH acknowledges funding from the Royal Society RPG-2016-186. T.S.L. and I.B. acknowledge the support of the US Department of Energy, Office of Basic Energy Sciences, Division of Materials Sciences and Engineering. Parts of this work was performed, at the Centre for Integrated Nano-technologies, an Office of Science User Facility operated for the US Department of Energy (DOE) Office of Science. We thank Philip Thomas for the ellipsometry measurement of the ITO thin film. Sandia National Laboratories is a multi-mission laboratory managed and operated by National Technology and Engineering Solutions of Sandia, LLC, a wholly owned subsidiary of Honeywell International, Inc., for the U.S. Department of Energy's National Nuclear Security Administration under contract DE-NA0003525.

## Author contributions

J.B., E.H. and W.L.B. conceived the idea; J.B. designed and built the experiment with input from C.T. and S.H.; J.B. carried out all measurements; T.S.L. and I.B. grew and characterized the ITO films; J.B. and E.H. developed and carried out the data analysis; S.H. developed the dynamic model; J.B. and E.H. wrote the manuscript with input from all the authors.

## Competing interests

The authors declare no competing interests.
