## [Peer Review File · Nature Communications]

Reviewers' Comments:

Reviewer #1:

Remarks to the Author:

The manuscript by J. Bohn et al. reports all-optical switching of a resonant plasmon resonance supported by sub-100-nm-thick ITO thin film around its epsilon-near-zero wavelength. (Nearly) degenerate pump-probe measurements were performed and a reflection modulation from 1% up to 45% is demonstrated. A coherent mechanism is proposed to further enhance the reflection modulation enabled by the hitherto well-known electron-heating mechanism, which changes the plasma frequency of ITO film. The motivation behind the work is that a Kretschmann-Raether configuration is adopted to provide a resonance, the shift of which gives rise to the observed, large modulation. Although this topic is interesting, unfortunately I found the results not comparable to the state-of-the-art that would meet the standard for publication in Nature Communications.

(1) The main reason is that the plasma frequency manipulation of transparent conducting oxides (such as ITO) around its ENZ region has been well studied and reported in the past several years. While the change in reflection from 1% to 45% is large, it is not too surprising because the starting value (1%) is low and the pump fluence is high.

(2) The dynamics observed with 227 THz, which was associated with a change to the phase of reflection, can also arise from the fact that this wavelength is near or at the zero-crossing of the derivative of the linear resonance, so that its sign can be very sensitive: under this circumstance, the sign can change from being positive during the initial rise time to negative during the hot electron cooling (or vice versa), and eventually recover to the baseline.

(3) Also, the same Kretschmann configuration for ITO film employing a prism coupling had indeed been demonstrated in the seminal work from S. Franzen's group back in year 2006 (e.g., J. Appl. Phys. 100, 054905, and several related papers). The other notable observation of the coherent mechanism has not been thoroughly investigated and explained, other than the fact that it caused some additional plasma frequency shift.

Reviewer #2:

Remarks to the Author:

Manuscript Number: NCOMMS-20-27656-T

Title: All-Optical Switching of an Epsilon-Near-Zero Plasmon in ITO

The manuscript by Bohn et al. reports on the experimental observation and analysis of the nonlinear optical response of indium tin oxide resulting from the excitation of the so-called epsilon-near-zero (ENZ) plasmon. The authors conducted the experiment with a pump-probe arrangement using the Kretschmann configuration for the excitation of the ENZ plasmon. A maximum change in reflection of 45% was observed for degenerate pump/probe frequencies using TM polarization.

The subject of nonlinear optics with ENZ materials is quite relevant at the moment, as researchers try to understand better these materials and to identify their potential all-optical light control. I believe that this work would be of interest for researchers in the field.

The manuscript is well written and the experimental results and analysis provide solid evidence of the conclusions. I can recommend this manuscript for publication in Nature Communications, provided that the authors address the following points:

1. The authors claim in the conclusions that their results "could pave the way towards optical plasmon switching". However, it seems to me that a maximum change in reflection of 45% is far from the typical requirements for optical switching (typically 20 dB or more extinction ratio). Can the authors comment on this, and perhaps give a fair statement in the text in this regard?

2. Regarding the unusual dynamics in the orange experimental data in Fig. 2a, the authors mentioned the following: "to the best of our knowledge, have not previously been observed." Then a dynamic model is introduced that qualitatively reproduce the results. However, the authors do not attempt to explain the reason of the unusual dynamics. Since some considerable differences are observed between the theory and experiment, it would be important that the authors give a physical explanation of the phenomenon so that the experimental data is well supported.

3. At present, the authors only report on the intensity dependent reflectance of the structure. But many other studies report as well the intensity dependent refractive index (Δn). I consider important that the authors include information about Δn so that one can compare with values of Δn obtained in other experiments. I believe that this would be useful and appreciated by the community.

Reviewer #3:

Remarks to the Author:

To start, let me first congratulate the reviewers on the completion of their work and thank them for submitting their manuscript to Nature Communications. After reading, I find the main focus is on the intentional use of ENZ plasmon modes (e.g. anomalous dispersion region) in the ENZ thin film and the observation of their effect on the overall switching performance with related dynamics. The authors note different transient responses based on the selection of the probe wavelength (Fig. 2a), and also explore this as a function of polarization, angle of incidence, and pump-probe frequency combination. In the end, they illustrate strong switching performance (which is expected and well documented), but highlight an additional contribution that occurs when the pump and probe are both TM polarized and degenerate.

The general premise of the article makes sense, in that utilizing the ENZ plasmon mode can increase absorption of the pump which is critical to improving the nonlinearity, while also introducing a low Q resonance into the picture to aid the sensitivity of the pump. Both should lead to an improved nonlinear response, and has been suggested as future directions to improve the nonlinearity (for example see Ref. 24).

Although angular dependence (manuscript ref 36), polarization dependence (S Benis et al, SPIE Proc, 10916, 2019), and to some extent pump & probe wavelength dispersions (E. Giuseppe et al, OMEGA 8(11) 3392-3400, 2018, D. J. Hagan et al, SPIE Proc., 11080, 2019) have been explored in ENZ materials, they have largely been separate works with different materials, under different conditions, with fewer data points than illustrated here, and largely investigate these effects individually and not collectively. As a result the present paper collects and combines these responses and dependencies in a single place with a single film. This is useful. Moreover, I very much applaud the discussion around the dubious reference 36, which among the core ENZ NLO community is known to have issues, although difficult to prove.

However, as it stands, I am not convinced on one of the primary novelty points of the paper, the "coherent response" identified by the authors. This appears very much like an artifact of the dielectric prism/substrate, as I discuss below, and without careful examination and normalization of the nonlinearities in the prism/substrate itself this conclusion is questionable. Without this unique feature, the novelty of the work is reduced. Although the work still remains a very useful comprehensive reference point to the community, the novelty to warrant publication in Nature Communications is lacking in the opinion of this reviewer.

Below I provide questions/concerns on the manuscript that should be addressed before publication. Following I provide some additional comments/suggestions to the authors to consider.

1. The reason for the shape of the reflection curve noted in Fig. 2a (case II) is not clear and little discussion of this is provided. If I consider the reflection of the angle 227THz, 45deg (~30% unpumped) in fig 2c, the excitation shifts this the plasma frequency such that the reflection initially

decreases quite quickly (due to the increased absorption). Once this initial shift relaxes 227THz, 45 deg appears to be $\sim 0\%$ reflection. Following the system relaxes, and the reflection begins to increase from this low value back to its steady state value. This movement did not result in the reflection curve shown in figure 2a case II. one would need to modulate the plasma frequency such that a region of higher reflectivity (that is on the opposite side of the reflection minimum) was reached. Given the example in Fig. 2c, it doesn't appear this is the case as the minimum in R is more or less move onto the point of interest. Perhaps this is still occurring but the color scales on the figure do not provide sufficient detail to follow. If this is the case, I would suggest the authors to rethink the figure 2c to help to make the transient described in Fig. 2a case ii more clear (e.g. more resolution in the reflection axis). Again, I follow the logic on how it can occur, but given the magnitude of this change and regimes of pumping, this window appears quite small.

2. It is unclear why a coherent or "instantaneous/quasi-instantaneous" contribution should exist in the ENZ nonlinearity under the conditions illustrated. Although there will of course still be a "fast" component to the nonlinearity even for ENZ films (due to coherent motion of the electron population in the conduction band), the slow contribution should dominate under a degenerate case (see J. B. Khurgin et al, Fast and slow nonlinearities in ENZ materials, arxiv:2007.05613, 2020). Another way to put it is, the "fast contribution" even for ENZ materials is approximately the same magnitude as the "fast" component (virtual transitions) in dielectrics, and since the volume of the ENZ material is so small, its contribution should be negligible compared to the slow component which is generally several orders of magnitude stronger. In this case, the slow contribution would arise due to absorption of the pump and subsequent thermal relaxation that occurs on time scale much longer than is indicated by the temporal plots shown in Fig. 4c, where the coherent response is indicated. As a result, one should not expect to see a coherent term for the ENZ layer of the strength indicated in the results. This leads me to the next point.

3. Very little information is given on how the authors deal with nonlinear effects of the prism and index matching oil, and this is quite important. Due to the size of the prism with respect to the ENZ film, the pump/probe must propagate quite a distance through this material and would be reasonably well overlapped in space (small angular deviation between the beams). Thus, even though the nonlinearity of the prism is much lower, it may still contribute a large signal that can be comparable to (and perhaps even larger than) the ENZ response. As a result, how can the authors be sure that the coherent contribution mentioned in Fig. 4b is truly a response of the ENZ film itself and not occurring from the fast nonlinearity in the prism? Although the authors do utilize a chopper on the probe while presumably detecting it at the pump repetition rate to eliminate single beam effects, cross beam effects may still play a role (for example, two beam coupling that facilitates coherent energy exchange from pump to probe, facilitated by kerr index gratings generated in the prism, which would tend to increase the strength of the probe and appear as an increased reflectivity). Such effects also exhibit a strong polarization dependence, generally being eliminated (or greatly reduced) under cross-pol excitation, similar to the plasmon coupling that is illustrated, making them difficult to distinguish. A closer examination of Fig. 4c (probe is $\sim 240\text{THz}$ so reflection change is nearly monotonically positive) may suggest this playing a role. For example, a clear bi-exponential response is observed this is not typical from the previous experiments in ENZ nonlinearities. In the "with CC" curve, the "fast" contribution appears to be close to being symmetric in time, which again is an indication of a polarization driven nonlinear component that should be very minimal in ENZ.

a. First the authors should perform identical experiments as described in the manuscript on the prism itself to clearly identify the response of this prism. This may require modification of the setup as critical angles and such will be different, so care must be taken to interpret these results correctly. If a non-negligible response is measured, the authors must clearly identify how they have accounted for this effect and properly removed it from the ENZ experiments.

b. The authors may be able to examine the interplay between the plasmon and any dielectric components by repeating the measurement in Fig. 4b under the other two polarization combinations not discussed, in particular the TE-TE polarization. If the response is a coherent effect in the dielectric prism, it is expected that the signal would return in this case with similar magnitude (since the fused silica is isotropic), while you would not see the coupling to the plasmon. Comparison of these plots, where the probe is always in TE, to the two shown in the manuscript will help answer these questions.

- c. In addition, the authors may consider to detect the transmitted probe (if not operating beyond the critical angle). This can provide additional information to understand energy conservation and help to identify if energy transfer from the pump has occurred.
- d. Lastly, some of the effects can be made worse by chirped pulses. It would be good for the authors to provide pulse characterization measurements (for example, FROG and beam profiles as available) to help readers understand the quality of their pump/probe pulses and if any other non-ideal effects may come into play.

In addition, I have a few questions/comments for the authors.

1. It would be helpful in Fig. 1c for the authors to include the light-line of the prism to illustrate the in-plane momentum of the excitation coming into the system for the min/max of the angles considered later. The dynamics in the following sections can be readily described by considering this picture, i.e. the slope of the prism light line for various angles and where it intersects the plasmon dispersion (defining pump/probe coupling to the plasmon) along with the sensitivity of this intersection point to shifts in plasma frequency (sensitivity of the probe to the modulation). I found myself constantly looking back at this plot to examine the results (as it is a very familiar plot to many) and a bit more explanation here, accompanied by the excellent Fig. 2a plots would be really helpful for solidifying the physics.

a. In figure S1, it is somewhat unclear why the peak in absorption does not experience much variation in frequency for a given angle. Can the authors comment on this? Perhaps it is because the index does not strongly vary within the region considered?

b. It would also be helpful if the authors would include a companion figure in S1 for the absorption of the pump in the TM case. This can accompany the modifications to Fig. 1c. This is somewhat understood from Fig. 2c, but at higher frequencies transmission dispersion may also play a role. So to make it clear, a direct plot of Absorption would be a nice reference to see.

2. With so many parameters changing, frequencies, polarization, angles, it is sometimes difficult to understand the conditions of the experimental conditions that produce a certain plot and they are not always listed. For example, on page 2 first paragraph of "Nonlinear near ENZ resonance" the authors describe Fig. 2a. The pump frequency/polarization was not mentioned. I can recommend that the authors develop a notation to illustrate in each case the values of various parameters, which ones are constant, what parameters are varying. For example; (pm: TE, 240 THz, 0deg; pr: TM, 210-260 THz, 45 deg). Although in some cases repetitive, it ensures that while the reader flips back and forth between the text description and the figure, one can readily recall the configuration that is being used to properly interpret the results.

3. The authors note that the unique temporal shape in Fig 2a case ii has not been noted before in literature. While I agree that it perhaps has not been focused on, it has been observed. For example, see M. Clerici et al, Nat. Commun, 8, 15839, 2017, Fig. 1. In this case there is a small deviation of the intraband reflection that goes below zero that is very comparable to the results obtained from the dynamic model (Fig. 2b dynamical model 236 THz curve). Although this was done at normal incidence, a similar feature can of course exist in this case as well, just centered around some reflectivity R_o as opposed to zero. Again, the statement is okay, but I would focus on the fact that it actually has been seen but it has not been discussed. We saw it in many measurements but avoided it because we didn't fully understand it at the time.

The manuscript was very well written in general so good work on this.

Reviewer #1

The manuscript by J. Bohn et al. reports all-optical switching of a resonant plasmon resonance supported by sub-100-nm-thick ITO thin film around its epsilon-near-zero wavelength. (Nearly) degenerate pump-probe measurements were performed and a reflection modulation from 1% up to 45% is demonstrated. A coherent mechanism is proposed to further enhance the reflection modulation enabled by the hitherto well-known electron-heating mechanism, which changes the plasma frequency of ITO film. The motivation behind the work is that a Kretschmann-Raether configuration is adopted to provide a resonance, the shift of which gives rise to the observed, large modulation. Although this topic is interesting, unfortunately I found the results not comparable to the state-of-the-art that would meet the standard for publication in Nature Communications.

We thank this reviewer for their time invested in analysing our work and hope that we can highlight the impact of our results below.

(1) The main reason is that the plasma frequency manipulation of transparent conducting oxides (such as ITO) around its ENZ region has been well studied and reported in the past several years. While the change in reflection from 1% to 45% is large, it is not too surprising because the starting value (1%) is low and the pump fluence is high.

The reviewer rightfully points out that there have been several works on ITO or ENZ materials in general. However, there are several novel aspects to our study:

- We are the first to employ the Kretschmann configuration in a pump probe schematic to study the nonlinear optical properties of an ENZ sample.
- We are the first to record and explain transient dynamics that result from a dynamically shifted resonance frequency (seen for 227THz in Fig 2a) [also see Question (2)].
- We are the first to highlight the existence of a coherent contribution that does not fit into the current picture of a homogenous thermal heating of the electron bath [also see Question (3)]. This coherent contribution is particularly important to point out as a lot of studies in the field are degenerate in frequency and polarization, potentially misunderstanding the contribution as just an additional thermal shift of the bulk plasmon frequency.
- Finally, we report large absolute reflection changes of ~45%, as also noted by this reviewer, which are achieved by engineering a system that has a low reflection to begin with.

We point out that our design is chosen precisely for this reason: to provide a stronger absorption (~1% reflection, ~99% absorption), and to improve upon the high fluence requirements that have been reported high for previous studies on unpatterned ENZ films. We also point out that the low

starting reflection is important for many applications, especially in telecoms, where the extinction ratio between an “ON” and “OFF” state is among the most crucial parameter. While we acknowledge that the fluences in our studies are still high, our experiments show a much stronger switching response for lower or similar fluences as used in other studies of ENZ layer systems suitable for usage in the telecom range, such as ITO [Science, 352(6287), 795–797. (2016)] or AZO [Optica, 2(7), 616. (2015); Physical Review Letters, 116(23), 233901 (2016); Optica, 7(3), 226. (2020)]. Moreover, there is considerable scope for further reducing the fluence required via material choice or more elaborate layer designs: In our study, we are looking at the switching capability of a very small volume of material in a layer of ITO that is just 60nm thick.

(2) The dynamics observed with 227 THz, which was associated with a change to the phase of reflection, can also arise from the fact that this wavelength is near or at the zero-crossing of the derivative of the linear resonance, so that its sign can be very sensitive: under this circumstance, the sign can change from being positive during the initial rise time to negative during the hot electron cooling (or vice versa), and eventually recover to the baseline.

There may be a misunderstanding here, as we believe our perception of the temporal signatures in the pump-probe dynamics matches that of the referee. To quote our submitted paper version: *“For 227 THz (orange, case II) the ENZ resonance shifts spectrally through the probe, and we observe some rather unusual dynamics in the pump-probe signal that, to the best of our knowledge, have not previously been observed.”* As suggested by the referee: the resonance first shifts to lower frequency due to heating, then returns to its equilibrium frequency due to subsequent cooling, giving rise to the multiple changes in the gradient. The “phase” we referred to in the previous manuscript was that of the reflection coefficient, which changes in sign as the resonance shifts through the probe frequency. However, we agree with the referee that our previous description was unnecessarily ambiguous. We have corrected this as follows:

*“For 227 THz (orange, case II) the ENZ resonance shifts spectrally through the probe, and we observe some rather unusual dynamics in the pump-probe signal that, to the best of our knowledge, have not previously been observed and discussed. **For this case, one initially observes a decrease in reflection, followed by an increase in reflection, as heating causes the resonance to shift through the probe frequency. Subsequent cooling then returns the resonance to its starting frequency. These combined effects cause multiple oscillations in the pump-probe dynamics. To better describe these effects, we have introduced a dynamic model, which captures effects arising due to rapid time-dependent changes to the reflection coefficient (details in supplementary S3).**”*

(3) Also, the same Kretschmann configuration for ITO film employing a prism coupling had indeed been demonstrated in the seminal work from S. Franzen's group back in year 2006 (e.g., J. Appl. Phys. 100, 054905, and several related papers). The other notable observation of the coherent mechanism has not been thoroughly investigated and explained, other than the fact that it caused some additional plasma frequency shift.

We thank the reviewer for the additional reference, which we now cite in the revised manuscript. As this reviewer notes, the Kretschmann configuration has been employed to study linear plasmonic properties of various thin-films, including for ITO. Indeed, our work is very much inspired by this early work which used this geometry to study the linear plasmon resonance in ITO. However, we are the first to employ this geometry to study the optical switching properties of ITO. This approach allows us to better demonstrate and characterise the nature and magnitude of the optical switching, and also gives us better insight into the physical mechanisms by e.g. separating out the thermal plasmon frequency shift from an additional coherent response.

We note that there is some misunderstanding of the proposed coherent response: we do not associate this with an increased plasmon frequency (as suggested by this reviewer). To address this misunderstanding, we have added a section to the supplementary information that describes our proposed mechanism.

Proposed coherent mechanism: "S5 Coherent contribution".

"When two coherent, co-polarised, near-degenerate beams impinge upon a plane, they generate an interference pattern. This gives rise to a spatially dependent change to the index of the material, which can cause diffraction generated signals in the experiment. The description we give below refers specifically to our pump-probe geometry, though it is important to realise that similar coherent signals (i.e. resulting from interference and diffraction) can result in many types of nonlinear optical measurement, even those employing a single focused beam [J. Opt. Soc. Am. B 20, 1290 (2003)].

For the geometry used in Fig. 4c, we model beams in glass (prism) with wavelength 1.25 μ m wavelength, probe angle = 48.3 deg and pump angle = 44.9 deg. These beams will result in the interference pattern inside the ITO, as shown in Fig. S6. For the beams used in our experiment ($I_{\text{pump}}=70 \text{ GW/cm}^2$ and $I_{\text{probe}}=0.13 \text{ GW/cm}^2$) we expect the interference pattern shown in Fig. S6b, with a spatially dependent oscillation in intensity of $\pm 4 \text{ GW/cm}^2$. Due to the intensity dependent index of refraction in ITO, one also expects a spatial dependence to the local index of refraction. Assuming the linear intensity dependence present in equation S4, one can expect the refractive index of the ITO layer to roughly resemble that shown in Fig. S6c.

Fig. S6 Diffraction through interference induced refractive index grating. **A**, Interference pattern of two equally intense beams (44.9 deg, 1250nm) and (48.3 deg, 1250nm) in glass. **B**, Interference pattern of 70 GW/cm² pump (44.9 deg, 1250nm) and 0.13 GW/cm² probe (48.3 deg, 1250nm) at glass/ITO interface. **C**, Corresponding refractive index distribution resulting from B, based on Eq. S4.

A spatial profile in the index of refraction will act as a diffraction grating, scattering pump light into the direction of the probe beam, and subsequently into our detector. While the spatial modulation in index is relatively small, leading to a relatively weak scattering effect, we only require a small intensity of the much stronger pump beam scattered in the direction of the probe to give a large switching signal. To obtain an estimate of this contribution, we use Comsol to calculate the diffraction pattern expected from the spatially varying index shown in Fig. S6c, assuming a 60 nm ITO layer with a uniform refractive index in z direction. This predicts a 1st order diffraction of ~0.1% of the pump beam that will be scattered in the direction of the probe beam. Hence, the signals we measure in our detector are not only based on the zero order reflection of the probe, but also a contribution of the first order diffraction of the pump. Considering the spatial overlap required for the pump to be scattered, we can directly estimate

the contribution of scattered pump to the “differential reflection” signal: this corresponds to $\Delta R_{CC} \sim \frac{0.1\% \cdot 70 \text{GW/cm}^2}{0.13 \text{GW/cm}^2} \sim 54\%$ of the probe intensity. This is even larger than the difference in peak TM and TE experimental signals, which is 15% (see Fig. 4c). However, due to their different temporal dynamics, the thermal and coherent contributions to the signal are expected to add sub-linearly. Moreover, we note that the refractive index relation used to calculate the scattering of the pump assumes full thermalisation, whereas the ITO will heat up during the evolution of the pump pulse. Thus the 54% predicted above is an overestimate, and we believe it to be consistent with the measured value of 15%.

For non-degenerate pump and probe, one expects a non-stationary interference pattern, i.e. one that changes quickly with time. For our geometry, one can easily show that effects of the grating will be washed out within the ~ 100 fs of our pulses when the pump and probe differ by only a few nm. Again, this is in agreement with our experiments presented in Fig. 4b, which show coherent signal only for near-degenerate measurements.

Reviewer #2:

The manuscript by Bohn et al. reports on the experimental observation and analysis of the nonlinear optical response of indium tin oxide resulting from the excitation of the so-called epsilon-near-zero (ENZ) plasmon. The authors conducted the experiment with a pump-probe arrangement using the Kretschmann configuration for the excitation of the ENZ plasmon. A maximum change in reflection of 45% was observed for degenerate pump/probe frequencies using TM polarization.

The subject of nonlinear optics with ENZ materials is quite relevant at the moment, as researchers try to understand better these materials and to identify their potential all-optical light control. I believe that this work would be of interest for researchers in the field.

The manuscript is well written, and the experimental results and analysis provide solid evidence of the conclusions. I can recommend this manuscript for publication in Nature Communications, provided that the authors address the following points:

We thank the reviewer for their kind words and are happy to address questions as follows:

1. The authors claim in the conclusions that their results "could pave the way towards optical plasmon switching". However, it seems to me that a maximum change in reflection of 45% is far from the typical requirements for optical switching (typically 20 dB or more extinction ratio). Can the authors comment on this, and perhaps give a fair statement in the text in this regard?

We thank the reviewer for this comment. Although we do not reach 20dB with this design just yet, we do find 45%/1% ~ 16.5dB a very promising value to build upon. We believe more carefully crafted layers could reduce the losses and thereby provide a larger upper value above 45%, while pushing the lower end below 1%. We also want to point out that our design works in a total internal reflection geometry, suitable for typical telecom environments such as fibers or integrated circuits. Moreover, there is considerable scope for further reducing the fluence required for shifting through material choice or more elaborate layer designs.

We have added statements to these effects in the conclusions, as per the referee's suggestion: "Considering the 16.5dB extinction ratio achieved in this proof of principle study, switching from near-perfect absorption to high reflection is useful from a signal processing point of view, and could pave the way towards optical plasmon switching at telecom frequencies. This could be further improved by more elaborate layer designs and/or further material improvements such as in carrier mobility."

2. Regarding the unusual dynamics in the orange experimental data in Fig. 2a, the authors mentioned the following: "to the best of our knowledge, have not previously been observed." Then a dynamic model is introduced that qualitatively reproduce the results. However, the authors do not attempt to explain the reason of the unusual dynamics. Since some considerable differences are observed between the theory and experiment, it would be important that the authors give a physical explanation of the phenomenon so that the experimental data is well supported.

We thank the referee for this constructive criticism, which has also been pointed out by the other reviewers. To address this, we have adjusted the paper as follows:

“For 227 THz (orange, case II) the ENZ resonance shifts spectrally through the probe, and we observe some rather unusual dynamics in the pump-probe signal that, to the best of our knowledge, have not previously been observed and discussed. For this case, one initially observes a decrease in reflection, followed by an increase in reflection, as heating causes the resonance to shift through the probe frequency. Subsequent cooling then returns the resonance to its starting frequency. These combined effects cause multiple oscillations in the pump-probe dynamics. To better describe these effects, we have introduced a dynamic model, which captures effects arising due to rapid time-dependent changes to the reflection coefficient (details in supplementary S3).”

3. At present, the authors only report on the intensity dependent reflectance of the structure. But many other studies report as well the intensity dependent refractive index (Δn). I consider important that the authors include information about Δn so that one can compare with values of Δn obtained in other experiments. I believe that this would be useful and appreciated by the community.

We thank for this suggestion and have added the following information to the supplementary section S3.1:

“To enable comparison to other studies using the refractive index and n_2 as the nonlinear optical parameters we provide the nonlinear refractive index depending on the absorbed intensity:

$$n(I_{abs}) = n_0 + n_2 * I_{abs} + O(I_{abs}^2)$$

with

$$n_0 = \sqrt{\epsilon_\infty - \frac{\omega_{p,0}^2}{\omega^2 + i\omega\gamma_0}}$$

$$n_2 = \frac{\omega_{p,0}^2}{2n_0} \frac{(2\omega^2\omega_{p,2} - i * \omega \gamma_0(\gamma_2 - 2\omega_{p,2}))}{(\omega^2 - i\omega\gamma_0)^2}$$

To give an example, n_2 takes the value of $(0.01 - i * 0.016)/(GW/cm^2)$ for our ITO sample case and 1250 nm wavelength. This equation may result in an approximate conversion for other ITO sample designs, but $\omega_{p,2} = 0.38\%/ (GW/cm^2)$ is expected to be different for materials with different band-structures such as AZO. Furthermore, to compare between different layer thicknesses and/or pulse lengths, one should also consider that the absorbed energy density is the expected scaling parameter of the thermal nonlinearity, not the absorbed intensity.”

Reviewer #3:

To start, let me first congratulate the reviewers on the completion of their work and thank them for submitting their manuscript to Nature Communications. After reading, I find the main focus is on the intentional use of ENZ plasmon modes (e.g. anomalous dispersion region) in the ENZ thin film and the observation of their effect on the overall switching performance with related dynamics. The authors note different transient responses based on the selection of the probe wavelength (Fig. 2a), and also explore this as a function of polarization, angle of incidence, and pump-probe frequency combination. In the end, they illustrate strong switching performance (which is expected and well documented), but highlight an additional contribution that occurs when the pump and probe are both TM polarized and degenerate.

The general premise of the article makes sense, in that utilizing the ENZ plasmon mode can increase absorption of the pump which is critical to improving the nonlinearity, while also introducing a low Q resonance into the picture to aid the sensitivity of the pump. Both should lead to an improved nonlinear response, and has been suggested as future directions to improve the nonlinearity (for example see Ref. 24).

Although angular dependence (manuscript ref 36), polarization dependence (S Benis et al, SPIE Proc, 10916, 2019), and to some extent pump & probe wavelength dispersions (E. Giuseppe et al, OMEGA 8(11) 3392-3400, 2018, D. J. Hagan et al, SPIE Proc., 11080, 2019) have been explored in ENZ materials, they have largely been separate works with different materials, under different conditions, with fewer data points than illustrated here, and largely investigate these effects individually and not collectively. As a result, the present paper collects and combines these responses and dependencies in a single place with a single film. This is useful. Moreover, I very much applaud the discussion around the dubious reference 36, which among the core ENZ NLO community is known to have issues, although difficult to prove.

However, as it stands, I am not convinced on one of the primary novelty points of the paper, the “coherent response” identified by the authors. This appears very much like an artifact of the dielectric prism/substrate, as I discuss below, and without careful examination and normalization of the nonlinearities in the prism/substrate itself this conclusion is questionable. Without this unique feature, the novelty of the work is reduced. Although the work still remains a very useful comprehensive reference point to the community, the novelty to warrant publication in Nature Communications is lacking in the opinion of this reviewer.

Below I provide questions/concerns on the manuscript that should be addressed before publication. Following I provide some additional comments/suggestions to the authors to consider.

We thank the reviewer for taking time to look at our manuscript in great detail and are happy to hear that our measurements and discussions are appreciated. We hope remaining concerns will be resolved with our following comments:

(1.1) The reason for the shape of the reflection curve noted in Fig. 2a (case II) is not clear and little discussion of this is provided. If I consider the reflection of the angle 227THz, 45deg (~30% unpumped) in fig 2c, the excitation shifts this the plasma frequency such that the reflection initially decreases quite quickly (due to the increased absorption). Once this initial shift relaxes 227THz, 45 deg appears to be ~ 0% reflection. Following the system relaxes, and the reflection begins to increase from this low value back to its steady state value. This movement did not result in the reflection curve shown in figure 2a case II. one would need to modulate the plasma frequency such that a region of higher reflectivity (that is on the opposite side of the reflection minimum) was reached.

The ambiguity in our previous description of this effect has been pointed out by all reviewers and we are thankful to the referee for making us aware of the insufficiently clear description in the previous manuscript. We believe we have addressed this problem with:

“For 227 THz (orange, case II) the ENZ resonance shifts spectrally through the probe, and we observe some rather unusual dynamics in the pump-probe signal that, to the best of our knowledge, have not previously been observed and discussed. For this case, one initially observes a decrease in reflection, followed by an increase in reflection, as heating causes the resonance to shift through the probe frequency. Subsequent cooling then returns the resonance to its starting frequency. These combined effects cause multiple oscillations in the pump-probe dynamics. To better describe these effects, we have introduced a dynamic model, which captures effects arising due to rapid time-dependent changes to the reflection coefficient (details in supplementary S3).”

It is important to note that this “simple” description does not fully account for the expected changes. Since the changes to the reflection coefficient occur on a similar timescale to the pulse width and period, we need to convolve the expected response with the measurement pulse, as we describe in section S3. Moreover, for a complete description, we would also need to describe the dynamical absorption of the pump within the pump pulse itself - such a complex description is far from trivial, and beyond the scope of the current manuscript.

(1.2) Given the example in Fig. 2c, it doesn't appear this is the case as the minimum in R is more or less move onto the point of interest. Perhaps this is still occurring but the colour scales on the figure do not provide sufficient detail to follow. If this is the case, I would suggest the authors to rethink the figure 2c to help to make the transient described in Fig. 2a case ii clearer (e.g. more resolution in the reflection axis). Again, I follow the logic on how it can occur, but given the magnitude of this change and regimes of pumping, this window appears quite small.

We believe there is some misunderstanding here. In Fig 2c, each data point corresponds to an entire pump-probe measurement trace, with the value plotted defined by $R_{\text{pumped}} = R_0 + \Delta R_{\text{max}} + \Delta R_{\text{min}}$ [see Fig 2a]. This allows us to condense information from many measurements into one colour plot in order to visualize the overall movement of the plasmon resonance.

We apologise that this was not sufficiently clear in the previous manuscript, and have added:

“...[Defining a maximal response in the oscillatory case is problematic - for simplicity, we define $R_{\text{pumped}} = R_0 + \Delta R_{\text{max}} + \Delta R_{\text{min}}$ for all measured time delay scans.] While not being able to resolve the transient features seen in Fig. 2b, this reduces the complex dynamical information contained in one complete time delay scan into two quantities of interest: the reflection before arrival of the pump (R_0) and a quantity describing the maximally changing reflection on photoexcitation (R_{pumped}). This allows us to condense information from many measurements into one colour plot.

By varying both the incident angle and the degenerate pump/probe frequency, one can observe in these colour plots a plasmon dispersion shift (Figure 2c)...”

(2) It is unclear why a coherent or “instantaneous/quasi-instantaneous” contribution should exist in the ENZ nonlinearity under the conditions illustrated. Although there will of course still be a “fast” component to the nonlinearity even for ENZ films (due to coherent motion of the electron population in the conduction band), the slow contribution should dominate under a degenerate case (see J. B. Khurgin et al, Fast and slow nonlinearities in ENZ materials, arxiv:2007.05613, 2020). Another way to put it is, the “fast contribution” even for ENZ materials is approximately the same magnitude as the “fast” component (virtual transitions) in dielectrics, and since the volume of the ENZ material is so small, its contribution should be negligible compared to the slow component which is generally several orders of magnitude stronger. In this case, the slow contribution would arise due to absorption of the pump and subsequent thermal relaxation that occurs on time scale much longer than is indicated by the temporal plots shown in Fig. 4c, where the coherent response is indicated. As a result, one should not expect to see a coherent term for the ENZ layer of the strength indicated in the results. This leads me to the next point.

The “coherence” we refer to is between the pump and probe beams, and results from using a coherent laser source for the measurement. We apologise for not describing this coherent effect in sufficient detail in the previous manuscript. We have now added an entirely new section to the supplementary information detailing the effect called “S5 Coherent contribution”.

(3) Very little information is given on how the authors deal with nonlinear effects of the prism and index matching oil, and this is quite important. Due to the size of the prism with respect to the ENZ film, the pump/probe must propagate quite a distance through this material and would be reasonably well overlapped in space (small angular deviation between the beams). Thus, even though the nonlinearity of the prism is much lower, it may still contribute a large signal that can be comparable to (and perhaps even larger than) the ENZ response. As a result, how can the authors be sure that the coherent contribution mentioned in Fig. 4b is truly a response of the ENZ film itself and not occurring from the fast nonlinearity in the prism? Although the authors do utilize a chopper on the probe while presumably detecting it at the pump repetition rate to eliminate single beam effects, cross beam effects may still play a role (for example, two beam coupling that facilitates coherent energy exchange from pump to probe, facilitated by Kerr index gratings generated in the prism, which would tend to increase the strength of the probe and appear as an increased reflectivity). Such effects also exhibit a strong polarization dependence, generally being eliminated (or greatly reduced) under cross-pol excitation, similar to the plasmon coupling that is illustrated, making them difficult to distinguish. A closer examination of Fig. 4c (probe is ~ 240 THz so reflection change is nearly monotonically positive) may suggest this playing a role. For example, a clear bi-exponential response is observed this is not typical from the previous experiments in ENZ nonlinearities. In the “with CC” curve, the “fast” contribution appears to be close to being symmetric in time, which again is an indication of a polarization driven nonlinear component that should be very minimal in ENZ.

a. First the authors should perform identical experiments as described in the manuscript on the prism itself to clearly identify the response of this prism. This may require modification of the setup as critical angles and such will be different, so care must be taken to interpret these results correctly. If a non-negligible response is measured, the authors must clearly identify how they have accounted for this effect and properly removed it from the ENZ experiments.

We have performed measurements at 45 degree, both beams TM polarized and both wavelength set to 1200nm. We first insert the ITO sample, align and measure the time delay scan (1). After that we exchange the sample for a cover slip and only adjust the tilt of the sample holder and the angle of the sample holder to maximize reflection and compensate changes due to slight changes in index matching fluid binding to the prism. Then we measure time delay scan (2). Finally, we change back to ITO, again, only adjusting tilt and angle to retrieve the maximum reflection (for 1.5 μ m, due to higher reflection). With the then measured scan (3) we show that the nonlinear response was perfectly reproduced. Importantly, the response of the coverslip alone (2) is considerably smaller:

We have checked the absence of signal for case (2) for other angles, and signals are always considerably smaller than when the ITO is present, hence we can unequivocally attribute the bulk of the signal to the ITO film. We also note that the magnitude of the coherent contribution observed in experiment can be quantitatively predicted, as described in the next supplementary section "S5 Coherent contribution"

b. The authors may be able to examine the interplay between the plasmon and any dielectric components by repeating the measurement in Fig. 4b under the other two polarization combinations not discussed, in particular the TE-TE polarization. If the response is a coherent effect in the dielectric prism, it is expected that the signal would return in this case with similar magnitude (since the fused silica is isotropic), while you would not see the coupling to the plasmon. Comparison of these plots, where the probe is always in TE, to the two shown in the manuscript will help answer these questions.

We thank the referee for this suggestion. For the TE-TE case we expect the thermal contribution to reflection changes to be negligible, which can be understood by the minimal frequency dependence in TE-absorption as seen in Fig. S1a compared to the resonant pump shown in Fig. S1b. However, the refractive index modulation is still expected to be similar (only slightly reduced due to lower pump absorption). This is indeed what we observe in the TE-TE experiment - the resulting coherent contribution can be seen as a 10% relative reflection increase when near degeneracy of pump and probe, and is significantly larger than the thermal effect in the same measurement:

To confirm the negligible thermal response for a TE probe we simulate the dynamic reflection equivalent to Fig. 2b, with the only change that the probe is now TE polarized (new section: “S4 Probe polarization dependence of the nonlinear effect”). We find that the expected reflection changes are below 1%:

c. In addition, the authors may consider to detect the transmitted probe (if not operating beyond the critical angle). This can provide additional information to understand energy conservation and help to identify if energy transfer from the pump has occurred.

Due to restrictions determined by the geometry of our setup, this is unfortunately not possible to do.

d. Lastly, some of the effects can be made worse by chirped pulses. It would be good for the authors to provide pulse characterization measurements (for example, FROG and beam profiles

as available) to help readers understand the quality of their pump/probe pulses and if any other non-ideal effects may come into play.

We agree that these would be useful characterisation measurements to perform. We do not have FROG characterisation equipment in our lab, and so are not able to present this data. We have included the autocorrelation of our pulse at 1200 nm using an APE pulseCheck autocorrelator, and find no strong indication of chirp like behaviour:

The measured pulse width is 107 ± 5 fs, with no significant change in pulse length over the range in wavelengths employed in this work.

We now use this measured pulse width in place of the previously estimated pulse length of ~ 100 fs (taken originally from our OPA specification). In the revised manuscript, we have used this measured pulse width in our modelling (e.g. which gives peak intensity ~ 70 GW/cm² instead of 75 GW/cm²). Note that all conclusions are unaffected by this change, which points towards a modest strengthening of our reported nonlinearity. We thank the referee for this very useful suggestion.

In addition, I have a few questions/comments for the authors.

1. It would be helpful in Fig. 1c for the authors to include the light-line of the prism to illustrate the in-plane momentum of the excitation coming into the system for the min/max of the angles considered later. The dynamics in the following sections can be readily described by considering this picture, i.e. the slope of the prism light line for various angles and where it intersects the plasmon dispersion (defining pump/probe coupling to the plasmon) along with the sensitivity of this intersection point to shifts in plasma frequency (sensitivity of the probe to the modulation). I found myself constantly looking back at this plot to examine the results (as it is a very familiar plot to many) and a bit more explanation here, accompanied by the excellent Fig. 2a plots would be really helpful for solidifying the physics.

We thank the reviewer for his suggestion and believe that the following figure is more suitable than our previous version of Fig 1c:

a. In figure S1, it is somewhat unclear why the peak in absorption does not experience much variation in frequency for a given angle. Can the authors comment on this? Perhaps it is because the index does not strongly vary within the region considered?

We would like to remind the reviewer that we plot the absorption of our TE polarized (pump) beam, which is much less angle and frequency dependent than the resonant TM excitation. Additionally, we plot the frequency dependence of both n and k :

b. It would also be helpful if the authors would include a companion figure in S1 for the absorption of the pump in the TM case. This can accompany the modifications to Fig. 1c. This is somewhat understood from Fig. 2c, but at higher frequencies transmission dispersion may also play a role. So, to make it clear, a direct plot of Absorption would be a nice reference to see.

We have updated Fig. S1 to also include the TM pump:

2. With so many parameters changing, frequencies, polarization, angles, it is sometimes difficult to understand the conditions of the experimental conditions that produce a certain plot and they are not always listed. For example, on page 2 first paragraph of “Nonlinear near ENZ resonance” the authors describe Fig. 2a. The pump frequency/polarization was not mentioned. I can recommend that the authors develop a notation to illustrate in each case the values of various parameters, which ones are constant, what parameters are varying. For example; (pm: TE, 240 THz, 0deg; pr: TM, 210-260 THz, 45 deg). Although in some cases repetitive, it ensures that while the reader flips back and forth between the text description and the figure, one can readily recall the configuration that is being used to properly interpret the results.

We thank the reviewer for the very useful suggestion on notation, which we implemented throughout the manuscript.

3. The authors note that the unique temporal shape in Fig 2a case ii has not been noted before in literature. While I agree that it perhaps has not been focused on, it has been observed. For example, see M. Clerici et al, Nat. Commun, 8, 15839, 2017, Fig. 1. In this case there is a small deviation of the intraband reflection that goes below zero that is very comparable to the results obtained from the dynamic model (Fig. 2b dynamical model 236 THz curve). Although this was done at normal incidence, a similar feature can of course exist in this case as well, just centered around some reflectivity R_o as opposed to zero. Again, the statement is okay, but I would focus on the fact that it actually has been seen but it has not been discussed. We saw it in many measurements but avoided it because we didn’t fully understand it at the time.

We thank the reviewer for his comment, and we adjusted the wording to:

“For 227 THz (orange, case II) the ENZ resonance shifts spectrally through the probe, and we observe some rather unusual dynamics in the pump-probe signal that, to the best of our knowledge, have not previously been observed **and discussed.**”

Reviewers' Comments:

Reviewer #1:

Remarks to the Author:

The changes made in the revised version of the manuscript, especially the added section on the coherent response clarifies some confusions, and I can now recommend the publication of the manuscript in Nature Communications. My last only comment is on the introduction part regarding the change in the effective mass due to the non-parabolic electron dispersion in transparent conducting oxides, for which the paper (10.1103/PhysRevLett.115.147401) should be acknowledged (and ideally referred to) as this was the first paper (5 years earlier than reference 19) on hot-electron induced plasma frequency modulation in TCOs.

Reviewer #3:

Remarks to the Author:

I thank the authors for taking great care and consideration when reviewing the comments from the referees. The diligence here was appreciated.

Following the suggestions, I believe the paper has been improved and is easier to understand. Perhaps the most important contribution added, in the eyes of this reviewer, was the section S5 on the coherent contribution, the TE-TE measurement, and the measurement of the system with the coverslip and without ITO. Together these solidify the measurement and concept of the coherent contribution which before was very much lacking. In the end, the effect observed is two-beam coupling, which makes great sense as this is a non-instantaneous nonlinearity and is well-known to occur in such systems (for example for free electrons in gasses and plasmas, JK Wahlstrand and HM Milchberg, Opt Lett 36, 3822, 2011; P Michel et al PRL 113, 205001, 2014). It has even recently been noted in ITO films (see J. Paul et al, arxiv 2008:12824, 2020). As pointed out, this effect has not been discussed in the published ENZ community so far, but will of course be present and is something that researchers should be aware of when performing measurements.

To conclude, the primary points of the paper are as follows (noted from the reply to reviewer 1): use of Kretschmann configuration, observe and explain transient dynamics from a shifted resonance frequency, highlight the coherent term, and large reflection. My prior hesitation was on the accuracy of the claims regarding the coherent term, but as noted, the new data/explanation provided have assuaged these concerns. As a result, the improvement made in the efficiency by coupling to plasmons (decent absolute change in R with reduced fluence) combined with the explanation of the coherent effect are, in the eyes of this reviewer, both important contributions to the field of nonlinearities in ENZ and are sufficient to warrant publication in Nature Communications. I have a few last comments that I wish the authors to address before publication, detailed below.

1. The coherent effect you have observed is two-beam coupling. Please call it so. Using a general "coherent effect" terminology could lead others to interpret the effect differently (i.e. perhaps as some kind of new effect) which can cause confusion.

2. Please include in the supplementary file:

- a. the autocorrelation measurement
- b. the TE-TE measurement
- c. the measurement of the coverslip and ITO.
- d. Brief descriptions as necessary.

Without these one cannot truly be clear on what your "coherent effect" is as was noted in my original replies. The measurements you included very much solidify the arguments made in the paper and support section S5. Without them, other readers may come to similar conclusions as I did at first. You can answer these directly by included this information, thereby elevating the trustworthiness of your paper (something that is sometimes a challenge in nonlinear optics literature).

Recommendation:

The authors claim they are the first to highlight the coherent contribution. This makes it appear as if this was not present before or as if it was a new effect. I don't think this was the intent, because of course it has been there all along and is a well-known effect, it was just not isolated before (as noted in the abstract) and may have been lumped into other effects unknowingly. It is important to make this distinction, to make it clear that the effect is not entirely a new effect, but that this known physical effect is present in ENZ materials and will modify experimental findings if not correctly identified and handled. You may simply argue that you have isolated the contribution from two-beam coupling so that the nonlinearity of the background ENZ material can be accurately analyzed.

Note: the replies to the rest of my prior questions and comments are acceptable.

REVIEWER COMMENTS

Reviewer #1 (Remarks to the Author):

The changes made in the revised version of the manuscript, especially the added section on the coherent response clarifies some confusions, and I can now recommend the publication of the manuscript in Nature Communications. My last only comment is on the introduction part regarding the change in the effective mass due to the non-parabolic electron dispersion in transparent conducting oxides, for which the paper (10.1103/PhysRevLett.115.147401) should be acknowledged (and ideally referred to) as this was the first paper (5 years earlier than reference 19) on hot-electron induced plasma frequency modulation in TCOs.

We thank the reviewer for their valid suggestion. The paper is now also cited as reference number 19.

New Reviewer (instead of #2):

Optical switching utilizing this scheme of transiently heating conduction electrons in ITO (or other counterparts such as AZO) relies on high-pulse-energy femtosecond laser amplifiers (with repetition rates on the order of kHz up to a few hundred kHz). As such, even if further improvements are made by more careful structure design and/or material perfection, the switching rates will hardly be faster than 1 MHz (as ultimately limited by the capability of laser amplifiers, and also by the ability of the material to dissipate heat). Even though the switching amplitude is large (16.5 dB) and the recovery time is fast (a few picosecond), the readers should be reminded (with a sentence or two in a revised manuscript) about this limitation on the switching rates.

We thank this reviewer for pointing out concerns about the feasibility of a holistic optical switching environment. The limitations of this are now clarified with the following statement:

“[Considering the 16.5 dB extinction ratio achieved in this proof of principle study, switching from near-perfect absorption to high reflection is useful from a signal processing point of view, and could pave the way towards optical plasmon switching at telecom frequencies]. While the switching performance of the layer and the requirement for high intensities [could be further improved by more elaborate layer designs and/or further material improvements such as in carrier mobility], achievable switching rates will be ultimately restricted by limitations in repetition rate of pulsed optical sources.”

Reviewer #3 (Remarks to the Author):

I thank the authors for taking great care and consideration when reviewing the comments from the referees. The diligence here was appreciated.

Following the suggestions, I believe the paper has been improved and is easier to understand.

Perhaps the most important contribution added, in the eyes of this reviewer, was the section S5 on the coherent contribution, the TE-TE measurement, and the measurement of the system with the coverslip and without ITO. Together these solidify the measurement and concept of the coherent contribution which before was very much lacking. In the end, the effect observed is two-beam coupling, which makes great sense as this is a non-instantaneous nonlinearity and is well-known to occur in such systems (for example for free electrons in gasses and plasmas, JK Wahlstrand and HM Milchberg, Opt Lett 36, 3822, 2011; P Michel et al PRL 113, 205001, 2014). It has even recently been noted in ITO films (see J. Paul et al, arxiv 2008:12824, 2020). As pointed out, this effect has not been

discussed in the published ENZ community so far, but will of course be present and is something that researchers should be aware of when performing measurements.

To conclude, the primary points of the paper are as follows (noted from the reply to reviewer 1): use of Kretschmann configuration, observe and explain transient dynamics from a shifted resonance frequency, highlight the coherent term, and large reflection. My prior hesitation was on the accuracy of the claims regarding the coherent term, but as noted, the new data/explanation provided have assuaged these concerns. As a result, the improvement made in the efficiency by coupling to plasmons (decent absolute change in R with reduced fluence) combined with the explanation of the coherent effect are, in the eyes of this reviewer, both important contributions to the field of nonlinearities in ENZ and are sufficient to warrant publication in Nature Communications. I have a few last comments that I wish the authors to address before publication, detailed below.

1. The coherent effect you have observed is two-beam coupling. Please call it so. Using a general “coherent effect” terminology could lead others to interpret the effect differently (i.e. perhaps as some kind of new effect) which can cause confusion.

The terminology has been clarified in text and as recommended later, we emphasized the isolation of the signal, rather than the discovery of it as seen in:

“For degenerate pump and probe frequencies, we highlight an additional two-beam coupling contribution, not previously isolated in ENZ nonlinear optics studies, which leads to an overall pump induced change in reflection from 1 % to 45 %.”

“Exclusively for the TM pump polarisation, we isolate an additional two-beam coupling (TBC) contribution.”

“For on-diagonal (degenerate) measurements, we can identify a noticeably larger reflection due to two-beam coupling (TBC). To our knowledge, this is the first time that a TBC contribution to the nonlinear signal has been isolated this clearly in a pump-probe measurement on ITO. We note that the TBC contribution may be concealed, and not readily separated, within other degenerate pump-probe measurements in the literature.”

“[a thermally driven switching process results in a shift in the plasmon resonance frequency of 20 THz for a pump intensity of 70 GW/cm²,] while additional two-beam coupling is observed for degenerate pump and probe frequencies.”

As well as in Figure 4:

2. Please include in the supplementary file:
 - a. the autocorrelation measurement
 - b. the TE-TE measurement
 - c. the measurement of the coverslip and ITO.
 - d. Brief descriptions as necessary.

Without these one cannot truly be clear on what your “coherent effect” is as was noted in my original replies. The measurements you included very much solidify the arguments made in the paper and support section S5. Without them, other readers may come to similar conclusions as I did at first. You can answer these directly by included this information, thereby elevating the trustworthiness of your paper (something that is sometimes a challenge in nonlinear optics literature).

We have added the sections to the supplementary as follows:

S6 TE-TE measurement

Due to the weak nonlinear response of the TE probe we expect only small thermal contributions to nonlinear measurements as discussed in section S4. However, if the pump is also TE polarised one expects a notable two-beam coupling contribution as the refractive index modulation inside the ITO layer remains similar to the case discussed in section S5. This is indeed what we observe in the TE-TE experiment as seen in Fig. S7. The resulting TBC can be seen as a 10% relative reflection increase when near degeneracy of pump and probe, and is significantly larger than the thermal effect in the same measurement.

Figure S7. TE-TE measurement. The absence of the thermal response for a TE probe leads to an isolated clear TBC feature when pumping with a TE polarized beam. (pr : TE, 45° , $f_{pr} = 240$ THz; pm : TE, $f_{pm} = 240, 260$ THz)

S7 Coverslip measurement

We have performed measurements at 45 degree, both beams TM polarized and both wavelength set to 1200 nm (see Fig. S8). We first insert the ITO sample, align and measure the time delay scan (1). After that we exchange the sample for a cover slip and only adjust the tilt of the sample holder and the angle of the sample holder to maximize reflection and compensate changes due to slight changes in index matching fluid binding to the prism. Then we measure time delay scan (2). Finally, we change back to ITO, again, only adjusting tilt and angle to retrieve the maximum reflection (for 1500 nm, due to higher reflection). With the then measured scan (3) we show that the nonlinear response was perfectly reproduced. Importantly, the response of the coverslip alone (2) is considerably smaller.

Figure S8. Coverslip measurement. To show the negligible nonlinear response of the prism, index matching fluid and prism we present a delay scan for the absence of ITO (2). To ensure the alignment not being influenced we take ITO measurements before and after with the same alignment procedure (1,3). We also show the electronic offset without a probe, corresponding to $R = 0$ %. (pr : TM, 45° , $f_{pr} = 250$ THz; pm : TM, $f_{pm} = 250$ THz)

We have checked the absence of signal for case (2) for other angles, and signals are always considerably smaller than when the ITO is present, hence we can unequivocally attribute the bulk of the signal to the ITO film.

S8 Autocorrelation measurement

The autocorrelation of our pulses at 1200nm was measured using an APE pulseCheck autocorrelator (see Fig. S9). The measured pulse length is 107 ± 5 fs, with no significant change in pulse length over the range in wavelengths employed in this work. The measurement does not indicate any chirp like behaviour.

Figure S9. Autocorrelation for pulse length and quality check. The autocorrelation was measured after the pulses passed through the prism with a wavelength of 1200 nm. The pulse has been measured multiple times, leading to a pulse time estimate of 107 ± 5 fs.

Recommendation:

The authors claim they are the first to highlight the coherent contribution. This makes it appear as if this was not present before or as if it was a new effect. I don't think this was the intent, because of course it has been there all along and is a well-known effect, it was just not isolated before (as noted in the abstract) and may have been lumped into other effects unknowingly. It is important to make this distinction, to make it clear that the effect is not entirely a new effect, but that this known physical effect is present in ENZ materials and will modify experimental findings if not correctly identified and handled. You may simply argue that you have isolated the contribution from two-beam coupling so that the nonlinearity of the background ENZ material can be accurately analyzed.

We implemented changes to highlight *the isolation of the signal, not its discovery*, as seen in the answer to the first remark of this reviewer.

Note: the replies to the rest of my prior questions and comments are acceptable.

Reviewers' Comments:

Reviewer #3:

Remarks to the Author:

All of my comments have been adequately addressed through the revisions. Given the current status of the manuscript, I support publication.

Reviewer #3

All of my comments have been adequately addressed through the revisions. Given the current status of the manuscript, I support publication.

We thank this reviewer for their support and their time invested in analysing our work.